# Relationship between HDL Cholesterol Efflux Capacity, Calcium Coronary Artery Content, and Antibodies against ApolipoproteinA-1 in Obese and Healthy Subjects

**DOI:** 10.3390/jcm8081225

**Published:** 2019-08-15

**Authors:** Nicolas Vuilleumier, Sabrina Pagano, Fabrizio Montecucco, Alessandra Quercioli, Thomas H. Schindler, François Mach, Eleonora Cipollari, Nicoletta Ronda, Elda Favari

**Affiliations:** 1Division of Laboratory Medicine, Diagnostic Department, Geneva University Hospitals, 1211 Geneva, Switzerland; 2Division of Laboratory Medicine, Department of Medical Specialties, Faculty of Medicine, University of Geneva, 1206 Geneva, Switzerland; 3First Clinic of Internal Medicine, Department of Internal Medicine, University of Genoa, 6 viale Benedetto XV, 16132 Genoa, Italy; 4Ospedale Policlinico San Martino, Genoa, 10 Largo Benzi, 16132 Genoa, Italy; 5Centre of Excellence for Biomedical Research (CEBR), University of Genoa, 9 viale Benedetto XV, 16132 Genoa, Italy; 6Division of Cardiology, “SS. Antonio e Biagio e Cesare Arrigo” Hospital, 6 via Venezia 16, 15121 Alessandria, Italy; 7Division of Nuclear Medicine—Cardiovascular Section, Department of Radiology and Radiological Science, School of Medicine, Johns Hopkins University, JHOC 3225, 601 N. Caroline Street, Baltimore, MD 21287, USA; 8Division of Cardiology, Department of Medicine, Johns Hopkins University, Baltimore, MD 21287, USA; 9Division of Cardiology, Cardiology Center, Geneva University Hospital, 1211 Geneva, Switzerland; 10Department of Food and Drug, University of Parma, Parco Area delle Scienze, 43124 Parma 27/A, Italy

**Keywords:** cholesterol efflux capacity, coronary artery calcium score, obesity, anti-apoA-1 IgG, autoantibodies

## Abstract

Aims: To explore the associations between cholesterol efflux capacity (CEC), coronary artery calcium (CAC) score, Framingham risk score (FRS), and antibodies against apolipoproteinA-1 (anti-apoA-1 IgG) in healthy and obese subjects (OS). Methods and Results: ABCA1-, ABCG1-, passive diffusion (PD)-CEC and anti-apoA-1 IgG were measured in sera from 34 controls and 35 OS who underwent CAC score determination by chest computed tomography. Anti-apoA-1 IgG ability to modulate CEC and macrophage cholesterol content (MCC) was tested in vitro. Controls and OS displayed similar ABCG1-, ABCA1-, PD-CEC, CAC and FRS scores. Logistic regression analyses indicated that FRS was the only significant predictor of CAC lesion. Overall, anti-apoA-1 IgG were significantly correlated with ABCA1-CEC (*r* = 0.48, *p* < 0.0001), PD-CEC (*r* = −0.33, *p* = 0.004), and the CAC score (*r* = 0.37, *p* = 0.03). ABCA1-CEC was correlated with CAC score (*r* = 0.47, *p* = 0.004) and FRS (*r* = 0.18, *p* = 0.29), while PD-CEC was inversely associated with the same parameters (CAC: *r* = −0.46, *p* = 0.006; FRS: score *r* = −0.40, *p* = 0.01). None of these associations was replicated in healthy controls or after excluding anti-apoA-1 IgG seropositive subjects. In vitro, anti-apoA-1 IgG inhibited PD-CEC (*p* < 0.0001), increased ABCA1-CEC (*p* < 0.0001), and increased MCC (*p* < 0.0001). Conclusions: We report a paradoxical positive association between ABCA1-CEC and the CAC score, with the latter being inversely associated with PD in OS. Corroborating our clinical observations, anti-apoA-1 IgG enhanced ABCA1 while repressing PD-CEC, leading to MCC increase in vitro. These results indicate that anti-apoA-1 IgG have the potential to interfere with CEC and macrophage lipid metabolism, and may underpin paradoxical associations between ABCA1-CEC and cardiovascular risk.

## 1. Introduction

Lately, impaired high-density lipoprotein (HDL) cholesterol efflux capacity (CEC) from macrophages, involving mainly the adenosine triphosphate (ATP)-binding cassette (ABC) transporter A1 (ABCA1) pathway, has gained considerable interest as a promising biomarker of atherosclerosis and cardiovascular (CV)-related risk, both in primary and secondary prevention settings [1,2,3], and served as proof of concept that HDL functional measures may provide improved CV risk stratification over standard lipid profile and other traditional CV risk factors [1,2,3]. Nevertheless, inverse and paradoxical positive associations between CEC and incident CV events have been reported in high-risk populations [4], and patients with metabolic syndrome have been shown to have an increased CEC, possibly due to increase pre-β-HDL levels [5,6]. To further fuel this controversy, no associations were retrieved between CEC and coronary artery calcium (CAC) score, either in the general population or in rheumatoid arthritis patients [3,7]. These contrasting observations relating CEC with CV risk and CAC score might be influenced by disease state or oxidative stress and immune-mediated inflammation.

Among factors influencing immune-mediated inflammation and related to lipid metabolism, we and others focused our investigations on antibodies against apolipoprotein A-1 (anti-apoA-1 IgG), the major protein fraction of HDL regulating ABCA1 activity, as a biomarker of poor general and CV outcomes, both in primary and secondary prevention settings [8,9,10,11,12,13]. In parallel, in vitro and animal studies have shown that anti-apoA-1 IgG could promote inflammation, atherogenesis, myocardial necrosis, and mice death through interaction with the Toll-like receptor (TLR) 2, 4 and the cluster of differentiation 14 (CD14) complex [14,15,16,17,18]. Accordingly, we demonstrated in obese but otherwise healthy subjects that anti-apoA-1 IgG were independent predictors of coronary artery lesion upon chest computed tomography [19]. Several groups have shown that these autoantibodies could also promote atherogenesis by affecting HDL anti-oxidant properties [20,21,22,23,24], and recently anti-apoA-1 IgG against the c-terminal part of the protein was found to be inversely associated with fibroblasts CEC [25].

In this pilot study, we explored whether specific pathways of HDL CEC (ABCG1-mediated, ABCA1-mediated, passive diffusion) could be differentially associated with the CAC score in healthy obese and non-obese participants, and whether anti-apoA-1 IgG are associated with a specific HDL CEC pathway modification on macrophages.

## 2. Experimental Section

### 2.1. Study Population and Design

The current investigation follows as sub-analysis of a previous published study [19], including 48 non-obese subjects (BMI < 30 kg/m^2^) and 43 obese subjects (BMI ≥ 30 kg/m^2^), without known traditional CV risk factors (such as arterial hypertension, smoking, and diabetes mellitus, anamnestic notion of variant angina, family history of premature coronary artery disease), clinically patent cardiovascular/systemic disease, or CV treatment (statins, any cardiac or vasoactive medication). From these 91 Subjects, 14 samples were missing from non-obese participants and 8 samples were missing from obese subjects, leaving 69 subjects available for HDL CEC determination. As reported earlier, “before inclusion in the cardiac perfusion assessment test, study participants underwent a complete visit, including a physical examination, electrocardiogram, blood pressure measurements, and blood puncture in a fasting state. Following inclusion, each study participant underwent multidetector computed tomography (MDCT) and CAC assessment. The study was approved by the University Hospitals of Geneva Institutional Review Board (protocol number 07-183), and each participant signed the approved informed consent form. This study has been conducted in compliance with the Declaration of Helsinki [19].

### 2.2. Study Endpoints

Two predetermined endpoints were considered for this pilot study. The primary endpoint was to test the ability of CEC to predict the presence of any coronary artery calcification on chest computed tomography (CT) scan, as described below. The secondary endpoint was to explore the possible association between anti-apoA-1 IgG and CEC. All CT scan data were assessed by two senior cardiologists blinded to the participants’ biochemical data.

### 2.3. Assessment of Coronary Artery Calcification by Chest CTscan

In the first step of the perfusion assessment, a 64-slice multidetector computed tomography (MDCT) (64-sliceBiograph HiRez TruePoint Positron Emission Tomography/Computed Tomography (PET-CT) scanner, Siemens, Erlangen, Germany) was performed to determine the CAC score. The scanner was operated in the single slice mode with an image acquisition time of 100  ms and a Section thickness of 3  mm. Prospective electrocardiogram (ECG) triggering was done at 55% of the R-R interval. Contiguous slices to the apex of the heart were obtained. CAC was considered present if three or more contiguous pixels with a signal intensity of >130 Hounsfield Unit were identified. The size of the lesion was automatically calculated, and the CAC was scored using the Agatston algorithm. The CAC was computed across all lesions denoted within the left main, left anterior descending (LAD), left coronary circumflex (LCx), and right coronary artery (RCA), and the sum of all lesion scores yielded the total CAC score [26]. As recommended, we used the CAC scoring system in a binary fashion (CAC present or absent) [26,27]. Accordingly, any Agatston score above 0 was considered as a present CAC lesion [26,27].

### 2.4. Biochemical Analyses

A conventional lipid profile and high-sensitive C-Reactive Protein (hsCRP) profile were performed on routine autoanalyzers. Low-density lipoprotein (LDL) cholesterol levels were derived from the conventional Friedwald equation. Anti-apoA-1 IgG serum levels were measured as previously described [8,9,11,12,13]. Briefly, Maxi-Sorb plates (Nunc) were coated with purified, human-derived delipidated apoA-1 (20 μg/mL; 50 μL/well) for 1 h at 37 °C. After 3 washes with phosphate buffered saline (PBS)/2% bovine serum albumin (BSA; 100 μL/well), all wells were blocked for 1 h with 2% BSA at 37 °C. Samples were diluted 1 to 50 in PBS/2% BSA and incubated for 60 min. Additional patient samples at the same dilution were also added to an uncoated well to assess individual nonspecific binding. After 6 further washes, 50 μL/well of signal antibody (alkaline phosphatase-conjugated anti-human IgG; Sigma-Aldrich, Saint Louis, MO, USA) diluted 1:1000 in PBS/2% BSA solution were incubated for 1 h at 37 °C. After 6 more washes (150 μL/well) with PBS/2% BSA solution, the phosphatase substrate p-nitrophenyl phosphate disodium (50 μL/well; Sigma-Aldrich) dissolved in diethanolamine buffer (pH 9.8) was added. Each sample was tested in duplicate, and absorbance, determined as the optical density at 405 nm (OD405 nm), was measured after 20 min of incubation at 37 °C (FilterMax, Molecular Devices, San Jose, CA, USA). The corresponding nonspecific binding value was subtracted from the mean absorbance value for each sample. Anti-apoA-1 IgG seropositivity cutoff was predefined as previously validated and set at an optical density (OD) value of 0.6 and 37% of the positive control value, as described earlier [8,9,11,12,13]. OD values ranged from 0 to 1.3, and corresponding index values were between 0 and 84.9%.

Pre-β-HDL levels were measured using a quantitative enzyme-linked immunosorbent assay (ELISA) kit for pre-β-HDL detection in human plasma [28], following the manufacturer’s instruction (Sekisui Diagnostic, Darmstadt, Germany). The coefficient of variation was 3.1% to 5.3% for individual analytical runs, and 4.9% to 9.1% between different analytical runs.

### 2.5. Serum HDL Cholesterol Efflux Capacity (CEC)

All serum samples were stored at −80 °C. The aliquots were slowly defrosted in ice. Four cholesterol efflux pathways were evaluated in cell cultures: total CEC (in J774 mouse macrophages treated with 0.3 mm cAMP for 18 h to up-regulate ABCA1 expression) [1]; passive diffusion (PD)-CEC (in J774 mouse macrophages, basal conditions) [29]; ABCA1-CEC (as a difference between total efflux and PD) [30]; ABCG1-CEC (in hABCG1-expressing Chinese hamster ovary cells, CHO-K1, as efflux difference between hABCG1-expressing and parent CHO-K1 control cells) [31]. In all determinations, cells were labeled with [1,2-^3^H]-cholesterol in the presence of acyl CoA: cholesterol acyl tranferase (ACAT) inhibitor (2 µg/mL, Sandoz 58035) for 24 h. The efflux was promoted for 4 h (6 h for ABCG1-CEC) to 2% (*v*/*v*) serum samples. Serum HDL CEC was expressed as a percentage of the radioactivity released to the medium in 4 h (6 h for ABCG1-CEC) over the total radioactivity incorporated by cells. Control samples were run to confirm the responsiveness of cells. Background efflux, evaluated in the absence of acceptors, was subtracted from each samples value. Serum samples were determined in triplicate, while a standard pool of human serum from our laboratory (SN1) permitted correction for inter-assay variability and a second serum standard pool (SN2) was used to determine inter-assay variability [29].

### 2.6. Antibody Anti-apoA-1 Modulation of Cellular Cholesterol Efflux

The efflux process was evaluated in J774 cells as previously described. Following the ABCA1 upregulation with cyclic adenosine monophosphate (cAMP) we incubated cells, during the efflux time (4 h), with 25 µg/mL of anti-apoA-1 IgG. The efflux was promoted to either 10 µg/mL of apo-A-1 or 2% (*v*/*v*) of human serum standard [32].

### 2.7. Measurement of Intracellular Cholesterol Content

J774 cells were cultured in 10% fetal calf serum (FCS) in RPMI cell medium at 37 °C in 5% CO_2_. To perform the experiments, cells were seeded in 24-well plates at a density of 2 × 10^5^ cells/well for 24 h. Cells were exposed for 24 h to either 10% (*v*/*v*) whole serum and control IgG or anti-apoA-1 IgG 40 µg/mL. Cellular cholesterol content before and after serum exposure was measured as previously described [33]. Briefly, at the end of the experiment, cell monolayers were washed with Phosphat-Buffered Salin (PBS) and lysed in 0.5 mL of a 1% sodium cholate solution in water supplemented with 10 U/mL DNase. Cholesterol was than measured fluorimetrically using the Amplex Red Cholesterol Assay Kit (Molecular Probes, Eugene, OR, USA) as described by the manufacturer. The amount of cholesterol in each well was measured by comparison with a cholesterol standard curve included in each experiment. An aliquot of the cell lysates was also taken to measure cell protein by a modified Lowry method [34]. Cell cholesterol content after exposure of cells to serum and immunoglobulin G (IgG) was expressed as µg of cholesterol /mg protein.

### 2.8. Antibody Anti-apoA-1 Modulation of Membrane Free Cholesterol: Assay of Cholesterol Oxidase

Cholesterol oxidase treatment was essentially as previously described [35]. Briefly, cells were labelled with 3 μCi/mL [3H] cholesterol. Cholesterol oxidase (1 U/mL) was added, and cells were incubated for 4 h. Lipid was extracted with isopropanol, and radioactive cholesterol and cholestenone were separated using thin-layer chromatography and quantified.

### 2.9. Anti-apoA-1 IgG Modulation of Cellular Cholesterol Esterification: ACAT Activity

Cells were treated as previously described for the measurement of intracellular cholesterol content. Cholesterol esterification was evaluated as the incorporation of radioactivity into cellular cholesteryl esters after addition of [14C]-oleate-albumin complex [36]. At the end of incubation, cells were washed with PBS and lipids were extracted with hexane/isopropanol (3:2). The extracted lipids were separated by thin layer chromatography (TLC) (isoctane/diethyl ether/acetic acid, 75:25:2, *v*/*v*/*v*). Cholesterol radioactivity in the spots was determined by counting liquid scintillation.

### 2.10. Measurement of Free Cholesterol Content in Cell Supernatant

Free cholesterol in supernatant was measured using the fluorometric method Amplex^®^ Red Cholesterol Assay Kit (Molecular Probe, Eugene, OR, USA) following the manufacturer’s instruction. The experiments were conducted without an ACAT inhibitor.

### 2.11. Statistical Analyses

Analyses were performed using Statistica software (StatSoft, Tulsa, OK, USA). Normally, distributed clinical data are presented as mean ± standard deviation (SD), whereas data following a non-normal distribution are presented as median and interquartile range (IQR). Accordingly, means were compared using bilateral student *T* test, and medians were compared with the Mann-Whitney-test. Proportions were compared using two-tailed exact Fischer test. Correlations between variables were assessed using Spearman test. Due to the limited sample size, adjusted analyses were limited to the different forms of cholesterol efflux analyzed and the 10-year Framingham risk score (FRS) for coronary heart disease risk prediction [37] (allowing the aggregation of all traditional CV risk factors within a single continuous variable). Cholesterol efflux and intracellular cholesterol data were expressed as mean ± SD. Treatment groups were compared using an unpaired two tailed Student’s *t*-test. Because of the explorative nature of this work with predefined endpoints, Bonferroni correction was not applied. Logistic regression analyses were used to assess associations between the presence of any CAC lesion and CEC, or anti-apoA-1 IgG as variables. In this model, the presence of any CAC lesion was set as the dependent variable. Associations are presented as odds ratios (OR) and corresponding 95% confidence intervals (95% CI). A p value below 0.05 was considered as statistically significant.

## 3. Results

### 3.1. Study Characteristics

Baseline demographic characteristics are presented in Table 1. As shown in this table, obese participants had lower levels of HDL, but higher triglycerides, hsCRP, and pre-βHDL levels when compared to non-obese subjects. No significant differences were found between control and obese subjects in terms of CEC mediated by ABCG1, ABCA1, or PD pathways and total (ABCA1 + PD) CEC, and no difference was observed between these two groups for the CAC score, the number of coronary artery lesions, or the presence of any lesion identified in chest CT scans. On the other hand, obese participants tended to have higher anti-apoA-1 IgG levels and a significantly higher anti-apoA-1 IgG positivity rate. Interestingly, anti-apoA-1 IgG positive participants (*n* = 6) were only observed in the obese subjects. No other significant association with any of the other parameters tested was retrieved (Table 1).

### 3.2. Associations between CEC Pathways, CAC Score, Framingham Risk Score, Pre-Beta-HDL, and hsCRP Levels

ABCG1 CEC was not found to be associated with CAC score or FRS, nor with pre-β-HDL levels in any of the study groups (Table 2). ABCA1 CEC was found to be positively associated with CAC score and the number of CAC lesions in overall subjects (*r* = 0.26, *p* = 0.02), and the strength of this association seemed to increase in obese subjects (*r* = 0.47, *p* = 0.004), whereas it was lost in non-obese participants (Table 2). The only significant association retrieved between CEC and pre-β-HDL levels was between pre-β-HDL and PD-CEC (*r* = 0.39, *p* = 0.02), which was only observed in obese subjects. Furthermore, FRS correlated positively with ABCA1-CEC in overall and control subjects, but not in obese participants, whereas we observed an inverse correlation between PD-CEC, CAC score, and FRS in overall and obese participants, but not in non-obese subjects (Table 2). HsCRP levels did not correlate with any forms of CEC in any subgroups (Table 2), nor with the FRS (*r* = 0.09; *p* = 0.43), with CAC score (*r* = −0.04; *p* = 0.75), or with the number of CAC lesions (*r* = −0.03; *p* = 0.77). This absence of correlation between hsCRP, FRS, the CAC score, and the CAC lesion was unchanged in obese and non-obese individuals (data not shown). Because of the significant correlation shown between CAC score and anti-apoA-1 IgG [19], as well as between CAC score/lesions and ABCA1- and PD-CEC, we performed a sensitivity analysis after excluding anti-apoA-1 IgG positive patients (*n* = 6) in the obese subgroup in order to see if excluding high anti-apoA-1 IgG values could change the associations between different forms of CEC with FRS, number of CAC lesions, and CAC score, as well as with pre-β-HDL. Due to the non-significant aforementioned associations with hsCRP levels and any forms of CEC, hsCRP was not considered in our sensitivity analyses. As shown in Table 2, after excluding anti-apoA-1 IgG seropositive subjects, the correlations between CAC score/lesion and ABCA1-CEC and PD-CEC remained significant and of the same order of magnitude, whereas the significant association between FRS and ABCA1 was lost. The correlation between FRS and PD-CEC was unchanged. This does not point to anti-apoA-1 IgG as a key determinant of the associations retrieved between CAC score, ABCA1-CEC, and PD-CEC. Finally, if logistic regression analyses indicated that ABCA1-CEC, PD-CEC, and FRS were all significant predictors of CAC lesion in univariate analysis, the FRS was found to be the only independent predictor of CAC lesion in the overall cohort (Table 3). Given the limited study sample size, these analyses were not performed in the obese and non-obese subgroups separately.

### 3.3. Associations between Anti-apoA-1 IgGs, Specific CEC Pathways, Pre-β-HDL, and hsCRP Levels

Spearman correlations showed a positive association between ABCA1-CEC and anti-apoA-1 IgG (*r* = 0.48, *p* < 0.001) and a negative association between PD-CEC and these antibodies (*r* = −0.33; *p* = 0.004; Table 4) on our overall participants samples. The anti-apoA-1 IgG association with ABCA1-CEC remained significant for the obese and non-obese subgroups. Nevertheless, if the association with PD-CEC was unchanged in non-obese subjects, it was lost in obese participants (Table 4). No associations were retrieved between pre-β-HDL serum levels and anti-apoA-1 IgG, nor between these antibodies and hsCRP (data not shown).

### 3.4. Anti-apoA-1 IgG-Mediated Modulation of Cellular Cholesterol Efflux

To further evaluate the possible impact of anti-apoA-1 IgG on CEC, we measured ABCA1-CEC and PD-CEC using standard serum or apoA-1 in J774 macrophages without cAMP stimulation in presence of anti-apoA-1 antibodies during the efflux time. As shown in Figure 1, cells incubated with the antibody showed an increased ABCA1-mediated efflux to the acceptors (4.28 ± 0.29 vs. 6.45 ± 0.45 *p* < 0.0001 to apoA1 as acceptor; 1.16 ± 0.15 vs. 1.81 ± 0.19 *p* < 0.0001 to human serum as acceptor). Conversely, the presence of the anti-apoA-1 IgG reduced the passive cellular efflux diffusion to both apoA-1 and human serum (0.77 ± 0.10 vs. 0.48 ± 0.11 *p* < 0.0001; 6.49 ± 0.05 vs. 4.67 ± 0.66 *p* < 0.0001).

### 3.5. Modulation of Anti-apoA-1 IgGs on Intracellular and Membrane Cholesterol Content

To understand the net effect of such dual modification of anti-apoA-1 IgGs on cellular cholesterol balance, we measured the impact of anti-apoA-1 IgGs on intracellular cholesterol content according to a previously validated method. As shown in Figure 2, anti-apoA-1 IgGs induced a significant increase in intracellular cholesterol content compared to control IgGs and baseline conditions.

To better investigate the possible anti-apoA-1 IgG-mediated passive diffusion inhibition, we look at the free cholesterol content in the membrane of J774 cells after four hour anti-apoA-1 IgG stimulation. As shown in Figure 3, anti-apoA-1 IgGs significantly reduced the free cholesterol membrane content when compared to the control IgGs or to the untreated condition. In the same experimental conditions, we observed a significant reduction of the free cholesterol content in the supernatant counterpart in the presence of anti-apoA-1 IgGs (Figure 4). Since ACAT is the key enzyme for the cholesterol esterification and its cellular redistribution, we measured the ACAT activity in the presence of anti-apoA-1 IgGs or control antibodies. As shown in Figure 5, anti-apoA-1 IgGs (and not the control IgGs) significantly increased ACAT activity.

Taken together, these results indicate that the presence of anti-apoA-1 IgG on J774 macrophages enhances ACAT activity leading to increased intracellular esterified cholesterol pools, and decreases membrane free-cholesterol content. Such decrease in membrane free-cholesterol content reduce the cholesterol gradient, leading to a decrease in passive diffusion PD followed by the reduction of free-cholesterol content in cell supernatants. In this context, the increase in ABCA1-efflux in the presence of anti-apoA-1 IgG can be interpreted as the consequence of increased intracellular lipid pools.

## 4. Discussion

In this study, we extend previous findings by demonstrating that on top of being significantly associated with higher CAC [19], anti-apoA-1 IgG levels are inversely associated with PD-CEC and positively with ABCA1-CEC in healthy obese subjects, despite the absence of ABCA1-CEC, ABCG1-CEC, and PD-CEC differences between obese and control participants. This apparently counterintuitive clinical observation derived from this small case-control study prompted us to evaluate the possible direct impact of anti-apoA-1 IgG on normal serum and apoA-1 CEC on macrophages, and on macrophage cholesterol content.

Corroborating our clinical observations, these in vitro investigations indicated that, per se, these autoantibodies could at the same time inhibit PD-CEC and potentiate ABCA1-CEC. PD efflux consists of the simple diffusion of free/unesterified cholesterol molecules across the membrane through the aqueous phase following their concentration gradient between the membrane and HDL/apoA-1, and accounts for the majority of total macrophage efflux (up to 80%) in normocholesterolemic conditions [38,39]. On the other hand, ABCA1-mediated efflux is an active pathway facilitating the transfer of unesterified cholesterol specifically to lipid-poor apoA-1, and represents up to 60% of total macrophage cholesterol efflux when activated by increased intracellular cholesterol load [38,39,40]. For this reason, determining the net effect of simultaneously blocking and activating these two important HDL CEC pathways at the cellular level was the key to providing a sound interpretation of our clinical and functional in vitro observations. As anti-apoA-1 IgG leads to increased intracellular cholesterol content in macrophages after 24 h of stimulation, we currently postulate that the anti-apoA-1 IgG overdriving mechanism consists of ACAT stimulation, leading to intracellular esterified cholesterol accumulation, decreasing the amount of membrane free-cholesterol available, and thus diminishing the cholesterol gradient requested for PD-CEC. In this context, the increase in ABCA1 efflux can be ascribed to a feedback loop aiming at preventing cellular cholesterol accumulation. Nevertheless, other possibilities cannot be formally excluded, and the exact mechanisms by which anti-apoA-1 IgG stimulates ACAT activity are still elusive. Several non-exclusive hypotheses are discussed below. Firstly, one can postulate an upstream mechanism, where anti-apoA-1 IgG could, per se, stimulate ACAT activity. With ACAT being an endoplasmic reticulum intracellular protein, an indirect effect may be evoked. Indeed, as ACAT deficiency has been shown to increase TLR4 expression and function in hepatocytes stellate cells [41], we cannot exclude the fact that anti-apoA-1 IgG-mediated TLR2/TLR4 stimulation may interfere with ACAT activity, although this hypothesis is currently devoid from of experimental evidence. Alternatively, as ACAT activation is known to occur in response to cholesterol overload [42], any factor increasing either the intracellular cholesterol synthesis or uptake could be susceptible to increase ACAT activity. Therefore, knowing whether anti-apoA-1 IgG could affect HGMCoA reductase or LDL-receptor expression and function is currently under investigation.

Alternatively, we can consider the anti-apoA-1 IgG-mediated ACAT stimulation as a downstream consequence of mechanisms related to PD-CEC inhibition. Given the complexity of PD-CEC, where desorption of free cholesterol molecules from the cell plasma membrane into the surrounding aqueous phase is the rate-limiting step [38,39,40], there are several possibilities, including those in direct relation with apoA-1 [40]. Firstly, by stabilizing lecithin-cholesterol acyltransferase (LCAT) activity, apoA-1 maintains a favourable cholesterol gradient. One could hypothesize that anti-apoA-1 IgG may decrease LCAT stability, attenuating the cholesterol gradient and followed by PD-CEC attenuation. Secondly, as apoA-1 can also facilitate the maintenance of such gradient, either by acting as an acceptor for cholesterol molecules diffusing away from the cell surface, or by mediating the process of membrane microsolubilization, we can also envisage an antibody-mediated interference of one of these acceptor-related processes [38]. Indeed, as c-terminal (c-ter) part of apoA-1 is known to mediate membrane microsolubilization important for PD-CEC to occur [43], and because anti-apoA-1 IgG in humans is biased against the c-ter part of apoA-1 [44], we cannot rule-out a direct and antagonistic effect of anti-apoA-1 IgG on PD-CEC. Finally, as mentioned above, the increased ABCA1-CEC in the presence of anti-apoA-1 IgG is likely to represent a homeostatic feedback loop. Nevertheless, as ABCA1 activity is regulated by apoA-1′s interaction with an epitope on the c-terminal part of apoA-1 [45], we cannot formally rule out a direct agonistic effect of these antibodies on ABCA1, or an indirect effect mediated through the antibody-induced macrophage interleukin-6 (IL-6) production known to increase ABCA1 efflux [46]. Despite these current mechanistic knowledge gaps requiring further detailed investigations, these results indicate that anti-apoA-1 IgG-induced ACAT stimulation followed by PD-CEC inhibition represent another mechanism by which these auto-antibodies can promote atherogenesis.

The second notable finding of the present study is the observation that in obese participants, macrophage ABCA1-CEC is positively associated with coronary atherosclerosis burden quantified by the CAC score, whereas no such association was retrieved in healthy controls. Although somehow contrasting with data published in primary and secondary prevention settings [1,2,3], these results are in line with previous observations in metabolic syndrome and rheumatoid arthritis [4,5,6], and lend weight to the notion that depending on disease-related systemic factors, the associations between macrophage ABCA1-CEC and coronary atherosclerosis burden are not necessarily straightforward. As metabolic syndrome patients share many pathophysiological similarities with obese patients, the existence of such a paradoxical association was suspected, but still remained to be demonstrated. Furthermore, providing PD-CEC results in parallel with ABCA1-CEC was found to be instrumental in facilitating the interpretation of these results. Indeed, the inverse association between CAC score and macrophage PD-CEC suggest that the lower the PD efflux, the higher the level of intracellular cholesterol, and thus atherosclerosis burden. In this context, the anti-apoA-1 IgG-dependent ABCA1-CEC increase, in fact, may not sufficiently compensate for the PD inhibition ascribed to anti-apoA-1 IgG. Furthermore, as our in vitro data demonstrated a substantial influence of anti-apoA-1 IgG on PD-CEC through ACAT stimulation, we performed a sensitivity analysis excluding anti-apoA-1 IgG seropositive subjects in the obese subgroup, which yielded similar results overall. Taken together, these results indicate that considering ABCA1-CEC without considering other concomitant functional effluxes may lead to erroneous conclusions. These results also suggest that the presence of anti-apoA-1 IgG may represent a new systemic factor susceptible to altering the usual association between ABCA1-CEC and CV risk.

This study has several limitations. The first one relates to the low number of subjects enrolled in this study, raising possible power issues when it comes to the plausibility of non-significant findings. Nevertheless, despite this power limitation, the significant findings derived from this cohort enabled us to generate hypotheses that were tested and validated in vitro, providing valuable insights to correctly interpret counterintuitive associations. A second limitation of this study is that we were not able to decipher the exact molecular mechanisms by which anti-apoA-1 IgG could stimulate ACAT activity and inhibit PD-CEC, and we did not explore their possible ability to modulate key cholesterol homeostasis regulating genes, which is currently under investigation (Pagano S. et al., under review). A further limitation is the fact that due to limited human sample availability, we could not evaluate the effects of human purified apoA-1 IgG in this study. Nevertheless, as we previously demonstrated that the commercial anti-human apoA-1 IgGs used in this study were promoting the same effects as human-purified IgG fraction containing high levels of these autoantibodies in vitro [13,18], and because these autoantibodies have been consistently used in different validated and published animal and in vitro studies [11,12,13,14,15,16,17,18,19], we expect similar impacts on in vitro lipid metabolism to occur by using human purified anti-apoA-1 IgG, even if not formally demonstrated. Finally, being focused on atherogenesis and foam cell formation, we did not evaluate whether the anti-apoA-1 IgG effects on lipid metabolism could be reproduced by other cell types, such as hepatocytes, and did not measure the scavenger B-I-related CEC.

## 5. Conclusions

In conclusions, the results of this study indicate that in healthy obese subjects there is a paradoxical positive association between CAC score and ABCA1-CEC, and a negative association between PD-CEC and CAC score. These associations could be partly explained by the propensity of anti-apoA-1 IgG to inhibit PD-mediated cholesterol efflux through ACAT stimulation and promote foam cell formation, notwithstanding an enhancing effect on ABCA1-mediated cholesterol efflux. Although preliminary, these results highlight the importance of simultaneously considering the different CEC pathways and using a translational approach for proper ABCA1-CEC interpretation.

## Figures and Tables

**Figure 1 jcm-08-01225-f001:**
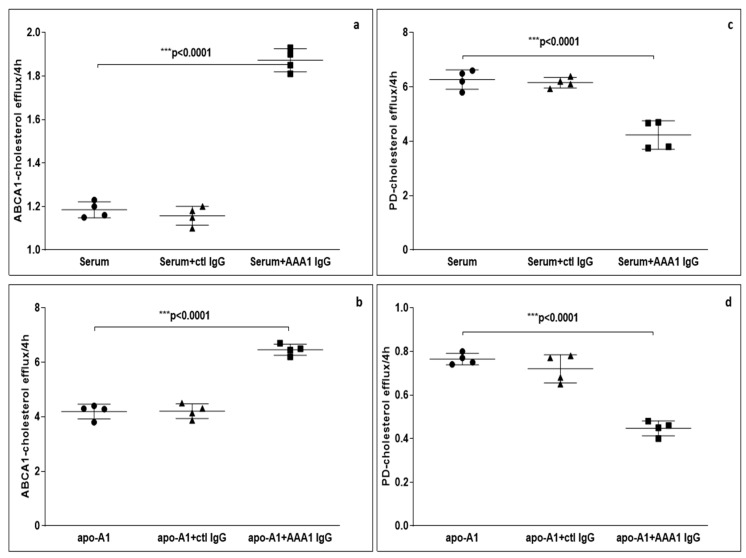
Cholesterol efflux from J774 to various acceptors in presence of anti-apoA-1 (AAA1) IgG. (**a**,**b**) The cells were activated with cAMP to measure the specific efflux from ABCA1 to serum or apoA-1 as acceptors after 4 h incubation. Data are expressed as the mean ± SD of the measurements done in triplicate and repeated four times (*n* = 4). Statistical differences were determined by one-way analysis of variance (one-way ANOVA), with *p* ≤ 0.05 being considered significant. (**c**,**d**) The cells were not activated in order to quantify the passive cellular efflux diffusion to both apoA-1 or human serum. The percentage of cholesterol efflux is shown. Data are expressed as the mean ± SD of the measurements done in triplicate and repeated four times (*n* = 4). Statistical differences were determined by one-way analysis of variance (one-way ANOVA), with *p* ≤ 0.05 being considered significant.

**Figure 2 jcm-08-01225-f002:**
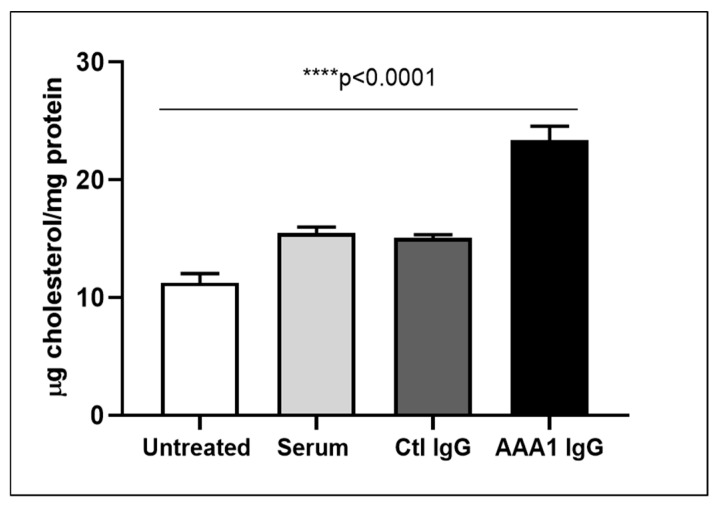
Anti-apoA-1 IgG increase the intracellular cholesterol content. Cell cholesterol content after exposure of cells to serum, anti-apoA-1 IgG, or control IgG, and was expressed as µg of cholesterol /mg total protein. Data are expressed as the mean ± SD of the measurements done in triplicate and repeated three times (*n* = 3). Statistical differences were determined by one-way analysis of variance (one-way ANOVA), with *p* ≤ 0.05 being considered significant.

**Figure 3 jcm-08-01225-f003:**
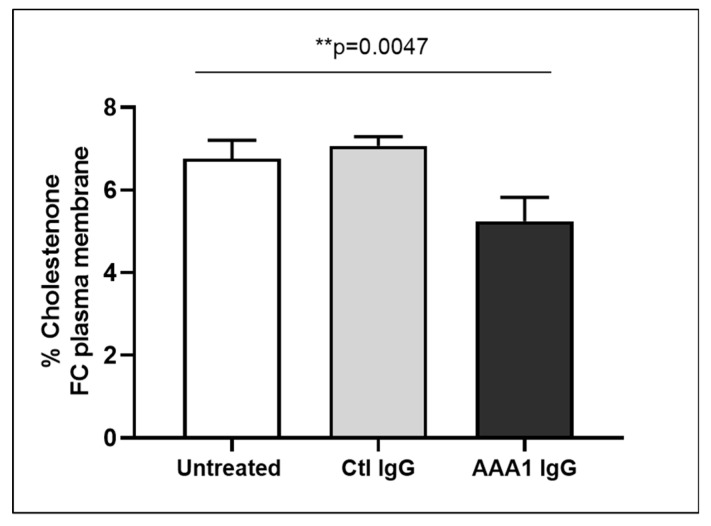
Free cholesterol in membrane. Monolayers were labeled with 3 μCi/mL [3H] cholesterol for 24 h in RPMI medium with 10% FCS, control IgGs or anti-apoA-1 IgGs 40 µg/mL. Cells were then washed and incubated with 1 U/mL cholesterol oxidase enzyme in Dulbecco’s Phosphate Buffered Saline (DPBS) for 4 h at 37 °C. Data are expressed as the mean ± SD of the measurements done in triplicate and repeated three times (*n* = 3). Statistical differences were determined by one-way analysis of variance (one-way ANOVA), with *p* ≤ 0.05 being considered significant.

**Figure 4 jcm-08-01225-f004:**
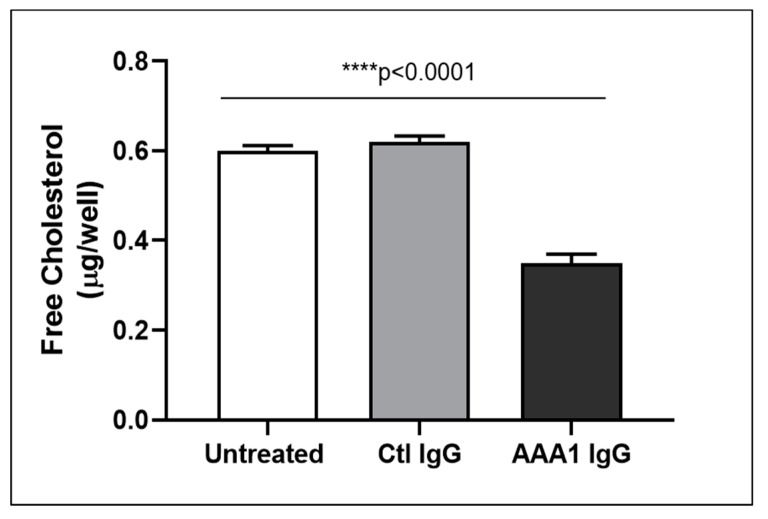
Free cholesterol in the cell supernatant. Free cholesterol content in J774 supernatant after exposure of cells to serum, anti-apoA-1 IgGs or control IgGs was expressed as µg of free cholesterol per well. Data are expressed as the mean ± SD of the measurements done in triplicate and repeated three times (*n* = 3). Statistical differences were determined by one-way analysis of variance (one-way ANOVA), with *p* ≤ 0.05 being considered significant.

**Figure 5 jcm-08-01225-f005:**
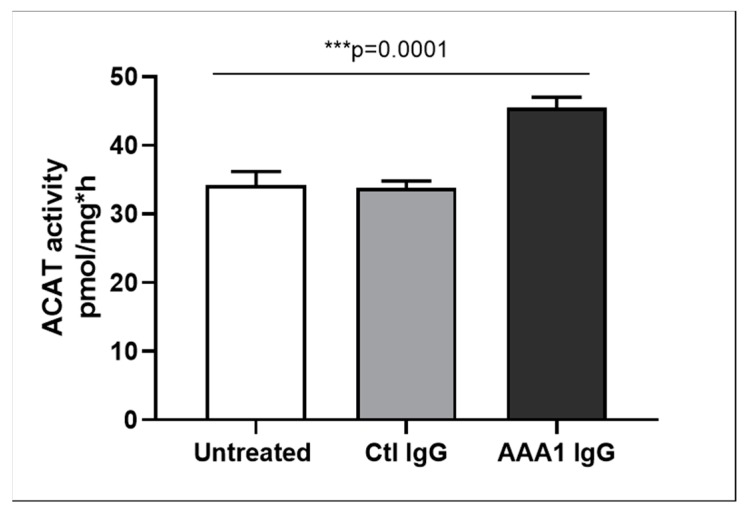
ACAT activity. Cells were incubated for 24 h in RPMI medium with 10% FCS, control IgG or anti-apoA-1 IgG 40 µg/mL. Monolayers underwent a second incubation (4 h) in the presence of [1-14C]-oleic acid albumin complex. Data are expressed as the mean ± SD of the measurements done in triplicate and repeated three times (*n* = 3). Statistical differences were determined by one-way analysis of variance (one-way ANOVA), with *p* ≤ 0.05 being considered significant.

**Table 1 jcm-08-01225-t001:** Baseline demographic and biological characteristics.

	*Overall* *(n = 69)*	*Obese* *(n = 35)*	*Non-Obese* *(n = 34)*	** p Value*
Age, mean (+/−SD)	44 (11.4)	44.6 (12.0)	43.5 (11.0)	0.61
Male Gender; n (%)	42 (60.8)	23 (65.7)	19 (55.8)	0.46
Weight in Kg, mean (+/−SD)	97.5 (26.37)	116 (19.3)	74.4 (12.5)	<0.001
Height in cm, mean (+/−SD)	74 (9.3)	173 (9.4)	172.2 (9.5)	0.56
BMI in kg/m^2^, mean (+/−SD)	32.2 (8.8)	39.0 (7.1)	25.1 (3.0)	<0.001
Framingham risk score; median (IQR; range)	1 (0.9–4; 0.9–16)	1 (0.9–6; 0.9–16)	1 (0.9–3; 0.9–15)	0.13
FRS > 10%; n (%)	7 (10.1)	6 (17)	1 (2.9)	0.10
***Chest CT:***				
Total CAC score, mean (+/−SD)	4.9 (15.8)	4.65 (14.8)	5 (17)	0.75
Number of CAC lesions, mean (+/−SD)	0.28 (0.75)	0.28 (0.71)	0.26 (0.79)	0.74
Presence of any CAC lesion; n (%)	10 (14.4)	6 (17.1)	4 (11.7)	0.73
***Lipid profile:***				
Total cholesterol in mg/dL, mean (+/−SD)	198.9 (36.4)	198.5 (41.4)	202.1 (31.5)	0.69
LDL cholesterol in mg/dL mean (+/−SD)	127.9 (32.2)	126.9 (36.9)	133.9 (26.7)	0.42
HDL cholesterol in mg/dL, mean (+/−SD)	47.1 (13.0)	42.1 (11.5)	50.4 (13.9)	0.04
Triglycerides in mg/dl, mean (+/−SD)	97.5 (73.2)	122.8 (70.8)	83.0 (67.0)	<0.001
Pre-β-HDL in μg/mL	42.6 (16.62)	49.2 (17.26)	35.6 (12.8)	<0.001
***Cholesterol Efflux Capacity:***				
ABCG1-mediated, mean (+/−SD)	4.04 (1.2)	4.04 (0.89)	3.85 (1.49)	0.96
ABCA1-mediated, mean (+/−SD)	3.98 (1.49)	4.18 (1.34)	3.67 (1.61)	0.18
Total, mean (+/-SD)	13.73 (1.55)	13.5 (1.57)	13.80 (1.57)	0.64
Passive diffusion, mean (+/−SD)	9.75 (2.05)	9.5 (1.9)	9.70 (2.16)	0.19
Anti-apoA-1 IgG OD; median (IQR; range)	0.31 (0.18–0.43; 0–1.3)	0.33 (0.2-0.48; 0.1–1.3)	0.26 (0.16–0.38; 0–0.56)	0.05
Anti-apoA-1 positivity, *n* (%) hsCRP, median (IQR; range)	6 (8.7) 2.9 (0.9–6; 0.9–26.1)	6 (17.1) 5 (2.5–8.7; 0.9–26.1)	0 (0) 1 (0.9–3; 0.32–11)	0.02 < 0.0001

Note: * *p* values were obtained by comparing obese versus non-obese subjects. For normally distributed parameters, *p* values were computed according to student *t*-test and for non-parametric parameters U-Mann Whitney test was used. Proportions were compared using two-tailed exact Fischer test. Abbreviations: SD = standard deviation; IQR = interquartile range; BMI = Body mass index; FRS = Framingham Risk Score; CT = Computed tomography; CAC = Coronary artery calcium; LDL = Low-density lipoprotein; HDL = High-density lipoprotein; ABCG1 = ATP-binding cassette subfamily G member 1; ABCA1 = ATP-binding cassette transporter 1; OD = Optical density; hsCRP = High-sensitivity C-reactive Protein.

**Table 2 jcm-08-01225-t002:** Correlations between CEC Pathways, CAC Score, Framingham Risk Score, Pre-Beta-HDL, and hsCRP Levels.

Overall Subjects (*n* = 69)			Obese Subjects (*n* = 35)		Non-Obese Subjects (*n* = 34)	
Correlations	*R* Value (Spearman)	*p* Value	*R* Value (Spearman)	*p* Value	*R* Value (Spearman)	*p* Value
**ABCG1-mediated CEC vs.:**						
CAC score	0.02	0.84	−0.03	0.82	0.10	0.57
Nr of CAC lesions	0.02	0.86	−0.04	0.81	0.09	0.61
Framingham RS	−0.01	0.90	0.06	0.70	−0.09	0.59
Pre-β-HDL	−0.22	0.09	−0.19	0.27	−0.23	0.18
hsCRP	0.007	0.94	0.04	0.77	0.22	0.20
**ABCA1-mediated CEC vs.:**						
CAC score	0.26	0.02	0.47	0.004	0.05	0.76
Nr of CAC lesions	0.26	0.02	0.47	0.004	0.04	0.80
Framingham RS	0.30	0.01	0.18	0.29	0.40	0.01
Pre-β-HDL	−0.08	0.48	−0.15	0.38	−0.25	0.15
hsCRP	0.05	0.44	0.04	0.79	−0.12	0.47
**Passive Diffusion vs.:**						
CAC score	−0.30	0.01	−0.46	0.006	−0.10	0.53
Nr of CAC lesions	−0.29	0.01	−0.45	0.006	−0.09	0.60
Framingham RS	−0.33	0.006	−0.40	0.01	−0.22	0.21
Pre-β-HDL	0.15	0.20	0.39	0.02	0.06	0.72
hsCRP	−0.01	0.90	0.06	0.63	0.17	0.35
**Correlations after excluding anti-apoA-1 IgG seropositive individuals in obese participants (*n* = 29)**						
**Correlations**	***R* Value (Spearman)**					***p* Value**
**ABCG1-mediated CEC vs.:**						
CAC score	0.01					0.72
Nr of CAC lesions	0.01					0.97
Framingham RS	0.06					0.72
Pre-β HDL	−0.13					0.49
**ABCA1-mediated CEC vs.:**						
CAC score	0.39					0.03
Nr of CAC lesions	0.39					0.03
Framingham RS	0.24					0.20
Pre-β HDL	−0.14					0.48
**Passive diffusion vs.:**						
CAC score	−0.39					0.04
Nr of CAC lesions	−0.39					0.04
Framingham RS	−0.45					0.009
Pre-β-HDL	0.41					0.03

**Table 3 jcm-08-01225-t003:** Logistic regression analyses for CAC lesion prediction

Univariate Analyses				Multivariate Analyses		
*Continuous Predictors*	*Odds Ratio*	*95% CI*	*p*	*Odds Ratio*	*95% CI*	*p*
ABCG1 CEC	1.10	0.64–1.91	0.72	1.02 *	0.51–2.05	0.94
ABCA1 CEC	1.17	1.17–3.13	0.009	1.38 **	0.67–2.86	0.38
Passive diffusion	0.56	0.32–0.97	0.03	0.78 ***	0.37–1.54	0.44
Framingham RS	1.31	1.13	0.004	1.27 ****	1.09–1.49	0.002

Note: * Adjusted for ABCA1 CEC, Passive diffusion and FRS; ** adjusted for ABCG1-CEC, passive diffusion, and FRS; *** adjusted for ABCG1-CEC and ABCA1-CEC, and FRS; **** adjusted for ABCA1, ABCG1, and passive diffusion.

**Table 4 jcm-08-01225-t004:** Spearman’s correlations between anti-apoA-1 IgGs, Specific CEC Pathways, Pre-β-HDL, and hsCRP levels

	Overall Subjects (*n* = 69)		Obese Subjects (*n* = 35)		Non-Obese Subjects (*n* = 34)	
Correlations	*R* value	*p* value	*R* value	*p* value	*R* value	*p* value
Anti-apoA-1 IgG (OD) vs.:						
CAC score	0.21	0.09	0.37	0.03	0.01	0.91
Nr of CAC lesions	0.21	0.09	0.37	0.03	0.02	0.88
Framingham Risk Score	−0.03	0.81	−0.03	0.82	−0.11	0.51
ABCG1-mediated CEC	−0.16	0.17	−0.23	0.18	−0.10	0.54
ABCA1-mediated CEC	0.48	0.00002	0.49	0.002	0.44	0.007
Passive diffusion	−0.33	0.004	−0.29	0.09	−0.36	0.03
Pre-β-HDL	0.17	0.89	−0.04	0.81	−0.06	0.9

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
