# Peer review of "Relationship between HDL Cholesterol Efflux Capacity, Calcium Coronary Artery Content, and Antibodies against ApolipoproteinA-1 in Obese and Healthy Subjects"

_jcm, 2019, doi:10.3390/jcm8081225_

Round 1
Reviewer 1 Report
Overall the study sounds technically consistent but to my opinion it needs some minor improvements:
In the study population and design section, authors state that form 91 subjects, 6 samples from none-obese participants plus 8 samples from obese subjects were missing, giving a final number of 69 included subjects (line 89). This is not true, either the final number was 77 or more patients were excluded. Please correct.
In the biochemical analysis line 122, what does CRP stand for? Please explain.
Why is that sometimes you present the data as mean (+,-SD) and sometimes as median (IQR)? To my understanding, one is used when applying parametric tests and the other for non-parametric tesst, I think it would be best to homogenize.
I think that the tables should not include the column for overall patients since you are studying the differences between obese and control subjects. It would make tables clearer. Also, consider presenting only what is significant specially for tables 2 and 3, there is no need to know Spearman R or Odds Ratio if not significant.
Lines 238. You talk about FRS without explaining the meaning nor the importance of calculating it. Please, introduce it somewhere.
Statistics is a bit confusing. Why do you apply a two way ANOVA analysis in figure 1? I think you should use a 1 way ANOVA. Also, in figure 5, why a t-student test? It should be a 1-way ANOVA. Please, make sure you are using the right test all the time.
Figure legends: figure 3 and 4 don't name statistical test used.
In figures 2, 3, 4 and 5, what does NT stand for?
Author Response
1.In the study population and design section, authors state that form 91 subjects, 6 samples from none-obese participants plus 8 samples from obese subjects were missing, giving a final number of 69 included subjects (line 89). This is not true, either the final number was 77 or more patients were excluded. Please correct.
We are thankful to the reviewer for spotting this inconsistency. It’s indeed 14 samples missing (and not 6 as previously stated) for the non-obese group, the number of missing sample the obese group is correct. We made the appropriate change in line 88 of the revised manuscript.
2.In the biochemical analysis line 122, what does CRP stand for? Please explain.
CRP stands for C-reactive protein. We added the full name in the text at line 125 then the abbreviation CRP.
3.Why is that sometimes you present the data as mean (+,-SD) and sometimes as median (IQR)? To my understanding, one is used when applying parametric tests and the other for non-parametric tesst, I think it would be best to homogenize.
When it comes to continuous clinical data, standard statistical practice implies that distribution of the values imposes that the choice depends on the data distribution. If the distribution is Gaussian or parametric (and you need to demonstrate it), expressing mean +/- SD is the first choice. If the distribution of the data set is non-parametric, median (IQR) must be used. The choice of the test to assess p values must also fit with the distribution of your data set. Using mean +/SD and a parametric test to compute p values in presence of non-normally distributed data is statistically incorrect and leads to inaccurate results and conclusions.
Accordingly and in the respect of these basic statistical principles, all the continuous variables summarized in our tables following a normal distribution were presented as mean +/- SD and compared with a two-sided t-test, whereas non-normally distributed data were presented as median (IQR) and compared using the standard non-parametric test (Mann-Whitney U test). We are of the opinion that not respecting this standard statistical practice will decrease the manuscript quality, reason why we feel that the results must be presented the way they are. We nevertheless provided further explanations to this respect in the statistical section lines 212-216 as follows: “Normally distributed clinical data are presented as mean ± standard deviation (SD), whereas data following a non-normal distribution are presented as median and interquartile range (IQR). Accordingly, means where compared using bilateral student T test, and medians were compared with the Mann-Whitney-test. Proportions were compared using two-tailed exact Fischer test.”
4.I think that the tables should not include the column for overall patients since you are studying the differences between obese and control subjects. It would make tables clearer. Also, consider presenting only what is significant specially for tables 2 and 3, there is no need to know Spearman R or Odds Ratio if not significant.
We are surprised by such demand standing in obvious opposition with most current guidelines (i.e. GRADE and CONSORT) regarding data reporting practice endorsed by the American College of Physicians. Indeed, these guidelines consider accurate presentation of results as the cornerstone of knowledge dissemination and an adequate mean to comply with the Helsinki declaration statement: “Authors have a duty to make publicly available the results of their research on human subjects and are accountable for the completeness and accuracy of their reports.” As such, we feel obliged to present our data in a summary of finding table accordingly, which intrinsically implies i) reporting precisely non-significant results, and ii) if any result is reported, exact p values must be reported rather than generalizing to >0.05 or mentioning “non-significant”. We are of the opinion that not respecting this standard data reporting practice will decrease the manuscript quality. We are convinced that the results must be presented the way they currently are, and hope that the reviewer will now share our point of view (additional information on this topic can be found following references: Boutron I et al. JAMA. 2010;303(20):2058-2064. Guyatt GH al. J Clin Epidemiol. 2013;66(2):158-72; Rosenbaum SER et al. J Clin Epidemiol 2010, 63:620-626. Lang TA et al ; How to Report Statistics in Medicine: Annotated Guidelines for Authors, Editors, and Reviewers Philadelphia American Coll of Physicians 1997 ; Altman DG et al.Statistics with Confidence: Confidence Intervals and Statistical Guidelines. 2nd ed London BMJ Books 2000;171:90).
Lines 238. You talk about FRS without explaining the meaning nor the importance of calculating it. Please, introduce it somewhere.
As requested, we provided details on FRS and the reason to use it in the statistical section (lines 219-222) as follows: “Due to the limited sample size adjusted analyzes were limited to the different forms of cholesterol efflux analyzed and the 10-year Framingham risk score (FRS) for coronary heart disease risk prediction, allowing the aggregation of all traditional CV risk factor within a single continuous variable,” and added the corresponding reference reference in the revised version of the manuscript: Wilson PW, D'Agostino RB, Levy D, Belanger AM, Silbershatz H, Kannel WB. Prediction of coronary heart disease using risk factor categories. Circulation. 1998;97(18):1837-47.
Statistics is a bit confusing. Why do you apply a two way ANOVA analysis in figure 1? I think you should use a 1 way ANOVA.
This was indeed an error and we thank the reviewer for drawing our attention to it. We modified the text it in the figure legend 1 to One-way ANOVA since we have applied a one-way ANOVA test (lines 322 and 326).
Also, in figure 5, why a t-student test? It should be a 1-way ANOVA. Please, make sure you are using the right test all the time.
Figure legends: figure 3 and 4 don't name statistical test used.
For the results reported in Figure 3 and 4 we have applied a T-test but as requested for homogenizing the statistical analysis for all the in vitro results we reanalyzed it applying a One-way ANOVA to all.
Figures 2-3-4-5 and their legends were then modified in line with the new p-values calculated.
In figures 2, 3, 4 and 5, what does NT stand for?
NT stands for non-treated but to make this point more clearly we replaced NT in the text of x-axis in figures 2-3-4-5 with “Untreated”.

Reviewer 2 Report
The current work examined in two groups of subjects (34 non-obese and 35 obese) without known traditional CVD risk factors correlations of coronary artery calcium (CAC) primarily with measures of three modes of cholesterol efflux capacity (ABCA1-CEC, ABCG1-CEC, and passive diffusion-CEC); and correlations of anti-apoA-I IgG with the three modes of CEC. In addition there were experiments on J774 macrophages assessing modulation by anti-apoA-I IgG of cellular cholesterol efflux as well as modulation by anti-apoA-I IgG of intracellular and membrane cholesterol contents. Main results demonstrated 1. significant positive correlation of CAC with ABCA1-CEC in obese but not in non-obese subjects and significant inverse correlation of CAC with passive diffusion-CEC again in obese but not in non-obese subjects; 2. significant correlation of anti-apoA-I IgG levels that were positive with CAC in obese subjects, positive with ABCA1-CEC in both subject groups, and negative with passive diffusion CEC in non-obese subjects; and 3. from macrophage experiments in response to anti-apoA-I IgG, increased ABCA1 mediated efflux and decreased passive diffusion mediated efflux; while intracellular cholesterol increased and membrane free cholesterol decreased.
The study addresses a significant issue regarding effects of anti-apoA-I antibodies on multiple modes of RCT, a key process in atherogenesis. As such, the work would be a worthy addition to this important area underlying differences in the modulation of separate RCT modes with the potential of leading to important clinical findings. A particular strength of the work is the approach of supporting initial results in human subjects with in-vitro macrophage experiments. Weaknesses in the work in addition to use of a surrogate marker of CVD events (CAC) are well covered in the limitations paragraph of the manuscript. Nevertheless, the study has potential merit if specific issues can be addressed focusing especially on clarity and accuracy in the presentation and specific points related to statistical modeling.
Specific comments
1. A careful revision of the manuscript is in order with regard to group label references in the text and tables. For example, in the Abstract (lines 35-37), correlation results apparently derived from Table 4 seem to be with regard to the overall subject group and not the obese group (OS) as stated. Additionally, why report the correlation of ABCA1-CEC with FRS having a p-value of 0.29 etc.?
2. The CAC score was used in a “binary fashion”; still it might be interesting to see histograms of the CAC scores separately for non-obese and obese subjects.
3. In section 2.4 (Biochemical analyses), it is mentioned that CRP was measured; yet it is never again referred to in the manuscript. If indeed measured, CRP results should be given especially in light of important associations between inflammation and obesity.
4. The sensitivity analysis (Section 3.2, lines 246 – 252) commentary is confusing. Apparently, the second portion of Table 2 refers to all subjects as indicated by the table column entry of n = 63. Would it not make sense to at least include separate analysis of the obese group given that it was only this group that had anti-apoA-I antibodies?
5. Regarding the logistic regression results of Table 3, since FRS apparently overwhelms any of the CEC results, would it not be useful to run a model with entry of the three CECs leaving out FRS to get a sense of potential individual contributions to CAC score. Additionally, a more logical adjustment to models would be obesity given the nature of the study.
Minor points
1. Fig 1 panels c and d: AD should be PD?
Author Response
Dear Editor,
Please, find here the revised version of the manuscript JCM-546099: “Relationship between HDL cholesterol efflux capacity, calcium coronary artery content and antibodies againstapolipoproteinA-1 in obese and healthy subjects”
We would like to thank you and the reviewers for the interest shown and for the constructive criticisms raised.
For the Reviewer 2 a document word is uploaded due to the presences of some supplementary tables and plots.

Round 2
Reviewer 2 Report
Concerns have been adequately addressed.